

# A practical application of reduced-copper antifouling paint in marine biological research

Andrea S. Jerabek[1,2], Kara R. Wall[2] and Christopher D. Stallings[2]

[1] College of Science, Northeastern University, Boston, MA, United States
[2] College of Marine Science, University of South Florida, St. Petersburg, FL, United States

## ABSTRACT

Biofouling of experimental cages and other field apparatuses can be problematic for scientists and has traditionally been addressed using frequent manual removal (e.g., scraping, scrubbing). Recent environmental restrictions and legislative changes have driven the development of less hazardous antifouling products, making antifouling paint a potential alternative option to manual removal. Consequently, the viability of using these newly developed products as a replacement for the manual cleaning of exclusion cages was experimentally investigated. There were six treatments tested, comprising three with settlement tiles in experimental cages coated with antifouling paint, two with settlement tiles in unpainted experimental cages, and one cage-free suspended tile. The three antifouling treatments comprised two reduced-copper paints (21% $Cu_2O$ and 40% $Cu_2O$) and one copper-free, Econea™-based paint (labeled "ecofriendly"). Antifouling paints were assessed for performance of preventing fouling of the cages and whether they elicited local effects on settlement tiles contained within them. All three paints performed well to reduce fouling of the cages during the initial six weeks of the experiment, but the efficacy of "ecofriendly" paint began to decrease during an extended deployment that lasted 14 weeks. The macro-community composition, biomass, and percent cover of settled organism on tiles within cages treated with copper-based paints (21% and 40% concentrations) were indistinguishable from tiles within the manually scrubbed cages. In contrast, settlement to tiles from the "ecofriendly" treatment was different in composition of macro-community and lower in biomass, suggesting the presence of local effects and therefore rendering it unsuitable for use in settlement experiments. The results of this study suggest that reduced-copper paints have the potential to serve as an alternative to manual maintenance, which may be useful for deployments in locations that are difficult to access on a frequent schedule.

Corresponding author
Andrea S. Jerabek,
andrea.jerabek1@gmail.com

## INTRODUCTION

Biofouling has historically been an implacable source of frustration for mariners (*Woods Hole Oceanographic Institution*, *1952*). Likewise, biofouling of gear used in field experiments (e.g., predator exclusion cages) can be of concern for scientists. The numerous negative effects of biofouling include reduced water flow, decreased oxygen levels, and increased weight and drag of infrastructures (*Fitridge et al.*, *2012*). Traditionally, scientific studies

have used manual techniques to remove biofouling, such as scrubbing and scraping (*Smith, Smith & Hunter*, *2001*; *Jompa & McCook*, *2002*; *Burkepile & Hay*, *2010*; *Burkholder et al.*, *2013*). While manual biofouling removal can be effective, it can also result in experimental disturbances, which could potentially skew study results (*Dobretsov, Williams & Thomason*, *2014*). Manual removal also requires frequent upkeep and monitoring (often at least twice per week). Consequently, it may be neither time- nor cost-effective for scientists to conduct experiments in locations where their infrastructures cannot be maintained on this frequent schedule.

One potential alternative to manual removal of biofouling organisms is the use of antifouling paints. Although antifouling paints were originally developed to prevent biofouling on vessel hulls, the technology is currently used in a multitude of commercial industries and research endeavors. For example, antifouling paints are frequently used by the aquaculture industry to eliminate the tremendous effort required to bring infrastructures on-shore for manual cleaning and maintenance (*Simpson, Spadaro & O'Brien*, *2013*). Additionally, antifouling paints are often used for docks, buoys, transducers, and site indicators. Currently, cuprous oxide is the most widely used antifouling biocide. However, there has been mounting evidence illustrating the negative environmental consequences of elevated copper levels on marine organisms (*Yebra, Kiil & Dam-Johansen*, *2004*; *Thomas & Brooks*, *2010*; *Guardiola et al.*, *2012*; *Qi et al.*, *2015*), resulting in the development of antifouling paints with lower copper concentrations. Additionally, recent legislative evaluations of copper-based paints have facilitated the emergence and acceptance of Econea$^{TM}$, an organic biocide, as the "ecofriendly" substitute to cuprous oxide. However, there is limited information regarding the toxicity and long-term environmental effects of Econea$^{TM}$ (*Holman et al.*, *2011*).

This study tested the efficacy of using the newly developed antifouling paints as an alternative to manually cleaning experimental apparatuses deployed *in situ*. Specifically, a field experiment was conducted to determine whether two levels of reduced-copper paints, and one copper-free, ecofriendly paint can be used for extended deployments of predator exclusion cages that house settlement tiles. Furthermore, antifouling paints were examined to determine if they elicited local effects on settlement to the tiles.

## MATERIAL AND METHODS

### Experimental design

To test the efficacy of using antifouling paints as an alternative to manual experimental cage maintenance, a field experiment was conducted to compare the macro-community composition, biomass accumulation, and percent cover of settlement tiles placed inside predator exclusion cages (Fig. 1). The experiment included six levels of a caging treatment: (1) "no-scrub" (no paint and no manual maintenance), (2) "scrub" (no paint, scrubbed clean twice weekly as in traditional manual maintenance), (3) "ecofriendly" paint (blue Hydrocoat Eco Ablative Antifouling Paint$^®$ treatment), (4) "21% Cu$_2$O" paint (blue "low-copper conc." CPP Ablative Antifouling Paint$^®$ treatment), (5) "40% Cu$_2$O" paint (blue "medium-copper conc." Horizons Ablative Antifouling Bottom Paint$^®$ treatment),

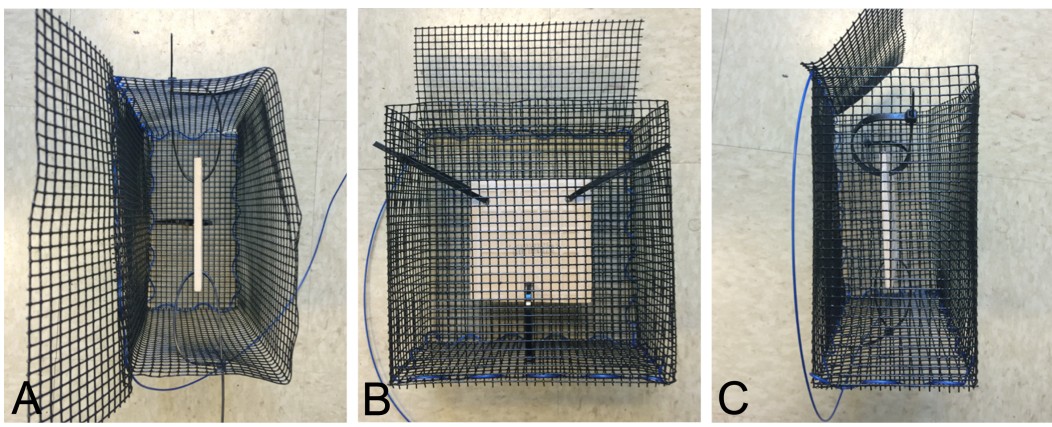

**Figure 1 Multiple views of an exclusion cage.** Photographs of an exclusion cage with a suspended settlement tile viewed from the top (A), front (B), and side (C).

and (6) "no-cage control" (suspended tile without a cage). Each painted cage received two hand-painted coats of a blue ablative antifouling paint, as instructed by the manufacturer, to ensure consistent thickness and to eliminate any potential experimental artifacts of color. The predator exclusion cages were 21.5 cm (l) ×10.6 cm (w) × 21.5 cm (h) and constructed of Vexar, a pre-galvanized, PVC-covered 0.064 cm metal mesh. The ceramic settlement tiles (11.5 cm (l) × 11.5 cm (w) × 0.6 cm (h)) were suspended within the cage via cable ties threaded through three 31.7 mm$^2$ drilled holes. Lost surface area due to drill holes was accounted for in percent cover calculations. A 5 cm buffer was maintained between the tile and cage frame to ascertain if antifouling paint ablation, or "wearing off," affected settlement to the tiles without direct contact. Sixty experimental units ($n = 10$ per treatment level) were deployed in Bayboro Harbor, St. Petersburg, Florida, USA on July 30th, 2015 (day 0) with a 0.5 m distance between cages to minimize cross contamination of antifouling paint. Cages were randomly dispersed within a grid design. The experiment concluded six weeks later on September 10th, 2015 (day 42). On day 42 of the primary experiment, all tiles received final measurements (see below), cages were scrubbed (where applicable according to assigned treatment level), and all cages were returned to the water where they remained untouched for an additional 62 days to qualitatively investigate the longer-term performance of the antifouling paints on the cage materials (i.e., 104 days total).

During the primary six-week experiment, tiles were removed from their cages once per week, weighed to measure biomass, and photographed for subsequent macro-community composition and percent cover analyses. Following data collection, tiles were promptly re-suspended within the cages and returned to the water. Initial tile masses were subtracted from all biomass measurements to obtain the weight of the fouling organisms on the tiles. Twice per week, the scrubbed cages were removed from the water for 15 min to provide time to be scrubbed with a plastic bristle brush. To control for the time the scrubbed cages were out of the water, all non-scrubbed cages were also removed from the water for 15 min,

thus ensuring uniformity across all treatments. Cages were inspected for performance throughout the duration of the experiment and photographed post-completion.

## Data analysis

Coral Point Count with Excel extensions (CPCe) version 4.1 software (*Kohler & Gill*, *2006*) was used to analyze tile photographs for macro-community composition. Each photograph was overlaid with 60 randomly stratified points. The organisms underneath each point were visually identified, after which the total number of points per species was divided by 60 to determine the percent cover of each organism per tile. The percent cover estimates of each species per tile were analyzed statistically to determine whether macro-community composition differed among treatments (See 'Statistical analysis'). Algae were clumped into a generalized "turf algae" functional group for macro-community composition analysis because photograph resolution was insufficient to distinguish among species.

The combined percent cover of all the fouling organisms on the tiles was analyzed using the software ImageJ (Version 1.48V). ImageJ was chosen because of its ability to the measure the area of atypical shapes on a photograph (i.e., macro-communities on a settlement tile). To accomplish this, the image type of each tile photograph was changed to 8-bit, creating a grayscale image. Following this transformation, the image threshold was adjusted to create a strong contrast between the settling organisms and the ceramic tile. The settled organisms were then visually distinct from the tile and the atypically shaped and discontinuous areas of the organisms on the tiles could be selected and quantified by ImageJ. The area of the settled organisms measured by ImageJ was then divided by the total relative area of the ceramic tile and multiplied by 100 to estimate overall percent cover of fouling organisms per tile.

## Statistical analysis

Macro-community composition analysis was completed via nonmetric multidimensional scaling (nMDS) in the statistical software program Primer using the Bray-Curtis (Sorensen) distance matrix. A PERMANOVA and subsequent pairwise comparisons were conducted on the percent cover data to determine whether macro-community composition differed among treatments. Statistical analyses of tile biomass and total percent cover were completed separately using the R statistical software programming environment (version 3.1.1). Tile biomass at the conclusion of the study (i.e., week six) was log transformed (*Sokal & Rohlf*, *1981*), and normality and homogeneity of the variance were confirmed via Shapiro–Wilk normality test and Levene's Test for homogeneity of variance, respectively. Separate one-way analyses of variances (ANOVA, alpha level $p = 0.05$) were conducted to test whether tile biomass and percent cover differed among treatments by day 42 (i.e., completion of the experiment). Pairwise Tukey HSD post-hoc tests were then conducted on significant main effects to test for differences in tile biomass and percent cover between treatment pairs.

## RESULTS

### Antifouling paint performance

Antifouling paint performance on the cages was consistent for the copper-based paints throughout the experiment. No more than three barnacle recruits were observed to be growing on the cages at the completion of the 42-day study. The ecofriendly paint had a similar antifouling performance with only few barnacles present on the cage frames. However, a thin algal slime was also observed on some of the cages treated with the ecofriendly paint. In contrast, unpainted, no-scrub cages quickly accumulated an algal slime (7–14 days), subsequently providing the foundation for macroalgae, barnacles, and other organisms to settle. The lack of scrubbing on the unpainted, no scrub cages, allowed for the uninterrupted development of fouling communities that likely inhibited flow through to the cages and affected settlement to the tiles. As expected, the unpainted, scrubbed cages remained free of fouling organisms due to the frequent maintenance performed throughout the study. After the additional two months of deployment (following the primary, six-week experiment), there was notable divergence in performance between the copper-based and ecofriendly antifouling paint treatments. Specifically, the copper-based paints had developed a thin algal slime that easily washed off. Conversely, the ecofriendly paint had a dense algal turf on the top of the cages accompanied by algal, barnacle, and hydroid growth on the cage sides and supporting zip ties.

### Macro-community composition on tiles

Over the six weeks of data collection, five different marine organism groups settled on the tiles: barnacles (*Amphibalanus amphitrite*; C Darwin), tubeworms (*Hydroides* spp.), oysters (*Crassostrea virginica*; JF Gmelin), mussels (*Perna viridis*; C Linneaus and *Geukensia granosissima*; GB Sowerby III), and various algal species (turf algae) (Table 1). The first organisms to colonize (barnacles, tubeworms, and turf algae) were present by day 7. Oyster recruits appeared on the tiles enclosed by the cages with no-scrub, scrub, and 40% $Cu_2O$ paint treatments by day 28, the ecofriendly and 21% $Cu_2O$ paint treatments by day 35, and the no-cage control by day 42 (Fig. 2). Mussel recruits were observed by day 21 on the no-scrub and ecofriendly treatments, but never accounted for more than 2% cover in any of the treatment levels throughout the experiment (Table 1). By day 42 barnacle percent cover was similar across treatments ranging from 80% to 100% cover. The ecofriendly painted and no-cage treatments had the largest percent covers of tubeworms and algae settled on the tiles. Percent cover of oysters was above 10% only in the no-scrub treatment, remaining below 5% in all other treatments (Fig. 2).

Percent cover was partitioned by species, and analyzed for the macro-community composition. By day 42, macro-community composition was similar for the scrub, 21% $Cu_2O$ paint, and 40% $Cu_2O$ paint treatments (PERMANOVA, $p > 0.05$; minimum convex polygon A in Fig. 3). However, the ecofriendly paint, no-scrub, and no-cage control treatments all formed distinct clusters in multivariate space (PERMANOVA, $p < 0.05$, minimum convex polygons B, C, D in Fig. 3, respectively).

Peer J

**Table 1 Species presence across treatments and day.** Species presence (indicated with "*") as a function of treatment and day of study. Treatment labels are: (1) no-scrub, (2) scrub, (3) ecofriendly, (4) 21% $Cu_2O$, (5) 40% $Cu_2O$, and (6) no-cage control. Mussels (*Perna viridis* and *Geukensia granosissima*) were indistinguishable within the first six weeks of the experiment.

| Treatment | Day 7 |   |   |   |   |   | Day 14 |   |   |   |   |   | Day 21 |   |   |   |   |   | Day 28 |   |   |   |   |   | Day 35 |   |   |   |   |   | Day 42 |   |   |   |   |   |
|---|---|---|---|---|---|---|---|---|---|---|---|---|---|---|---|---|---|---|---|---|---|---|---|---|---|---|---|---|---|---|---|---|---|---|---|---|
|  | 1 | 2 | 3 | 4 | 5 | 6 | 1 | 2 | 3 | 4 | 5 | 6 | 1 | 2 | 3 | 4 | 5 | 6 | 1 | 2 | 3 | 4 | 5 | 6 | 1 | 2 | 3 | 4 | 5 | 6 | 1 | 2 | 3 | 4 | 5 | 6 |
| *A. amphitrite* | * | * | * | * | * | * | * | * | * | * | * | * | * | * | * | * | * | * | * | * | * | * | * | * | * | * | * | * | * | * | * | * | * | * | * | * |
| *C. virginica* |   |   |   |   |   |   |   |   |   |   |   |   | * | * |   | * |   |   | * | * | * | * | * |   |   |   |   |   |   |   |   | * | * | * | * | * |
| *Hydroides* spp. | * | * |   |   |   |   | * | * | * | * | * | * | * | * | * | * | * | * | * | * | * | * | * | * | * | * | * | * | * |   | * | * | * | * | * | * |
| Mussel spp. |   |   |   |   |   |   |   |   |   |   |   |   |   | * |   | * |   |   |   |   |   |   |   |   |   | * | * |   |   |   |   | * | * | * | * | * |
| Turf algae | * | * | * | * | * | * | * | * | * | * | * | * | * | * | * | * | * | * | * | * | * | * | * | * | * | * | * | * | * | * | * | * | * | * | * | * |

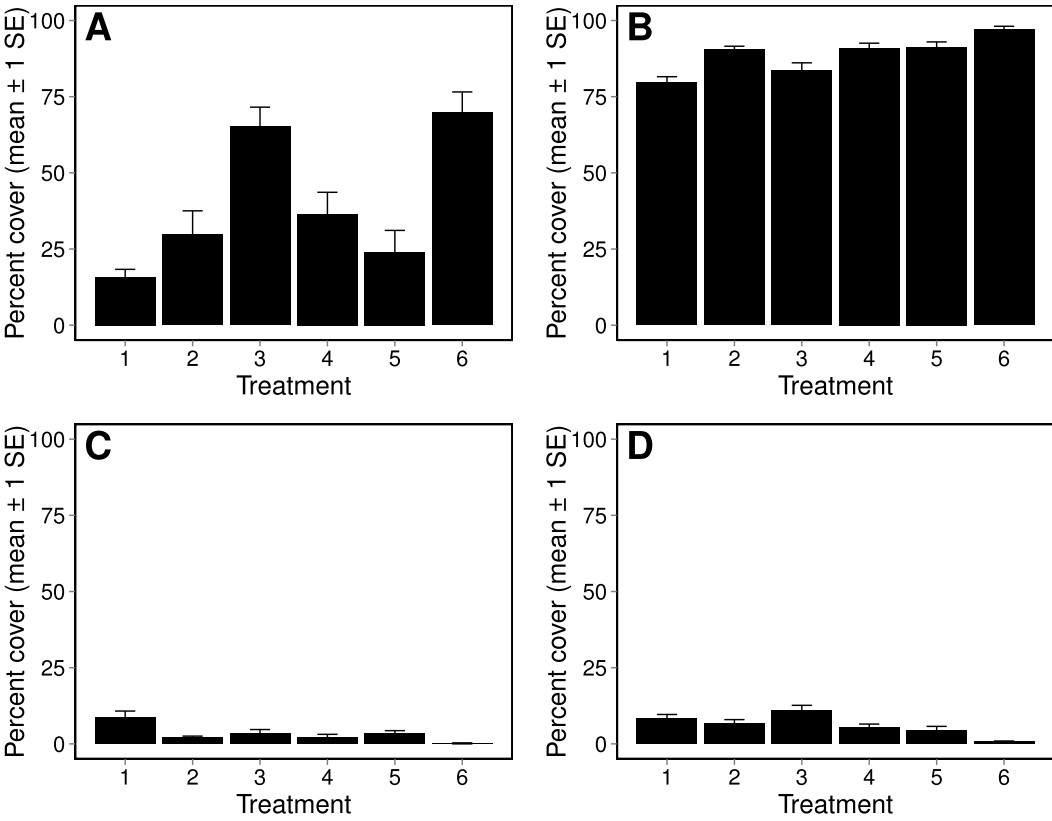

**Figure 2 Percent cover of recruited organisms on settlement tiles.** Percent cover (mean ± 1 SE) of the four primary recruited organisms on settlement tiles on day 42 across treatments. The primary recruited organisms consisted of turf algae (A), barnacles (B), oysters (C), and tubeworms (D). Treatment labels are: (1) no-scrub, (2) scrub, (3) ecofriendly, (4) 21% $Cu_2O$, (5) 40% $Cu_2O$, and (6) no-cage control.

## Biomass on tiles

By day 42 (i.e., completion of the experiment), there were distinct differences in biomass of settled organisms on the tiles across treatments (ANOVA, $F_{(5,54)} = 54.47$, $p < 0.001$, Fig. 4). Biomass of tiles from cages that were scrubbed and those coated with either 21% $Cu_2O$ paint or 40% $Cu_2O$ paint were not different from each other (Tukey HSD tests, $p > 0.05$), and accumulated the highest mean (SE) biomass of 480.2 g (±11.5; cluster A in Fig. 4). The average biomass on the tiles enclosed by the three aforementioned treatments was 37.7% higher than tiles enclosed by cages treated with ecofriendly paint (348.7 g ± 9.7; letter B in Fig. 4), 87.6% higher than the no-scrub treatment (255.9 g ± 7.1; letter C in Fig. 4), and 124.1% higher than the no-cage treatment (214.2 g ± 5.9; letter D in Fig. 4). The ecofriendly paint treatment on average had the next highest biomass accumulation, which was higher than both the no-scrub (Tukey HSD, $p < 0.001$) and no-cage treatments (Tukey HSD, $p < 0.001$). Biomass accumulation to the no-cage tiles was lower than that to no-scrub tiles (Tukey HSD, $p = 0.005$), presumably due to post-settlement predation on the un-caged tiles (e.g., herbivory, molluscivory).

Although the additional biomass accumulation on tiles was not quantified after the cages were left unmaintained for the 62 days following the six-week study, interesting
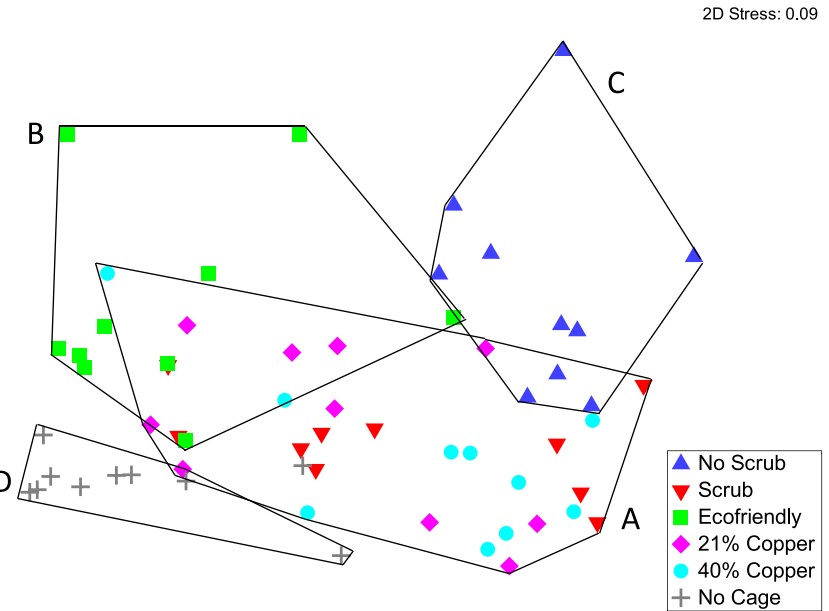

**Figure 3  Community composition of settlement tiles.** Nonmetric multidimensional scaling (nMDS) of community composition among the six experimental treatments. Minimum convex polygons and associated letter groupings (i.e., A, B, C, D) indicate differences among treatments according to the PERMANOVA and subsequent pairwise comparisons.

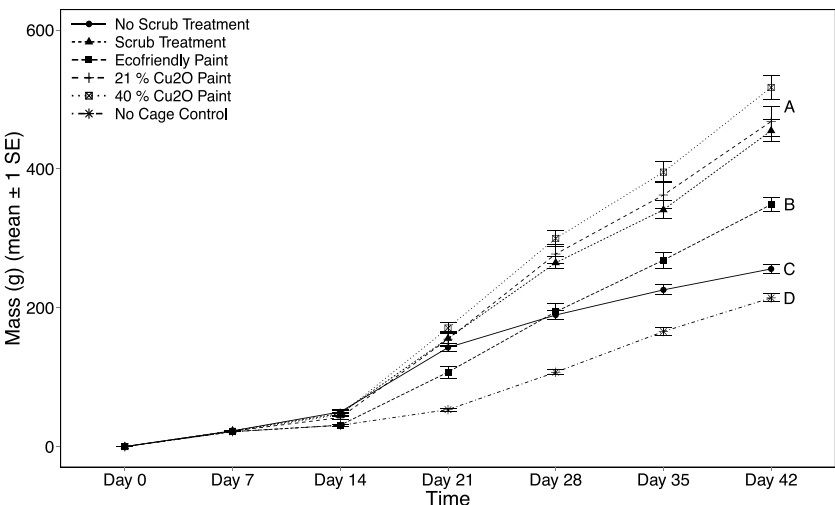

**Figure 4  Settlement tile biomass as a function of treatment and day.** Total biomass accumulation (mean $\pm$ 1 SE) on settlement tiles measured weekly for six weeks across the six experimental treatments. Significant differences among treatments according to Tukey HSD ($p < 0.05$) are indicated on the plot with different letters (i.e., A, B, C, D).

patterns were noted. For cages treated with copper-based paint, additional growth and settlement of organisms resulted in accumulation of growth that completely filled the cages, bringing the fauna in contact with the painted frame (Fig. 5). These cages were generally observed to be full of oysters that were dislodged from the tiles. Conversely, the organisms

in the ecofriendly painted cages remained constrained to the tiles, and the growth did not come into contact with the painted frames. In both unpainted treatments (i.e., scrub and no-scrub), barnacles were found growing on the tiles and cage frames themselves due to the absence of antifouling paint.

### Percent cover on tiles

All treatments except the ecofriendly paint and no-cage control had a mean percent cover of 80% by day 14 and 100% by day 21 (Fig. 6). The differences in percent cover of the ecofriendly and no-cage control treatments compared to the other four treatments emerged by day 14 of the study (ANOVA, $F_{(5,54)} = 9.15$, $p < 0.001$; Fig. 6). No significant difference of percent cover on tiles was observed between ecofriendly paint and the no-cage treatment (Tukey HSD, $p = 0.725$), but both were less than all other treatments (Tukey HSD, $p < 0.05$). By day 21, the ecofriendly paint treatment increased to 100% cover while the no-cage treatment remained as the only one without complete coverage of growth on the tiles (Fig. 6). The significant difference between the no-cage treatment and all other treatments remained through the completion of the experiment (Tukey HSD, $p < 0.05$). However, percent cover on tiles enclosed by unpainted cages that were not scrubbed had decreased by day 42, and were not significantly different than the no-cage tiles (Tukey HSD, $p = 0.21$).

## DISCUSSION

Using a controlled, six-week long field experiment, it was determined that the macro-community composition, biomass accumulation, and percent cover of settlement to tiles inside cages treated with reduced copper-based paints were indistinguishable from tiles inside unpainted cages that were scrubbed twice per week. Thus, there were no apparent local effects of the copper-based paints on biomass or macro-community composition of settled organisms, suggesting those paints could be a viable alternative for manual scrubbing of cages when conducting short-term settlement experiments. In contrast, settlement to tiles inside cages treated with the ecofriendly paint had significantly lower biomass and a different macro-community composition compared to the scrubbed cages. Thus, using the ecofriendly paint as an alternative to manual scrubbing in biological experiment apparatuses is not recommended.

The different macro-community structures and biomass of tiles inside the ecofriendly painted cages suggests that the paint altered species composition and decreased overall successful settlement on the tiles. While traditional copper-based ablative paints contain harsh solvents, Hydrocoat Eco® is a water-based ablative paint and free of harsh solvents. It is possible that the water in the ecofriendly antifouling paint caused the biocides to be dispersed differently than those in the copper-based antifouling paints, which may have accounted for the differences in recruitment, growth, and macro-community structure observed. It is also important to consider that in addition to Econea™, there are other chemicals found within the ecofriendly antifouling paint that may have contributed to the shift in macro-community composition and reduction of biomass rendering the paint

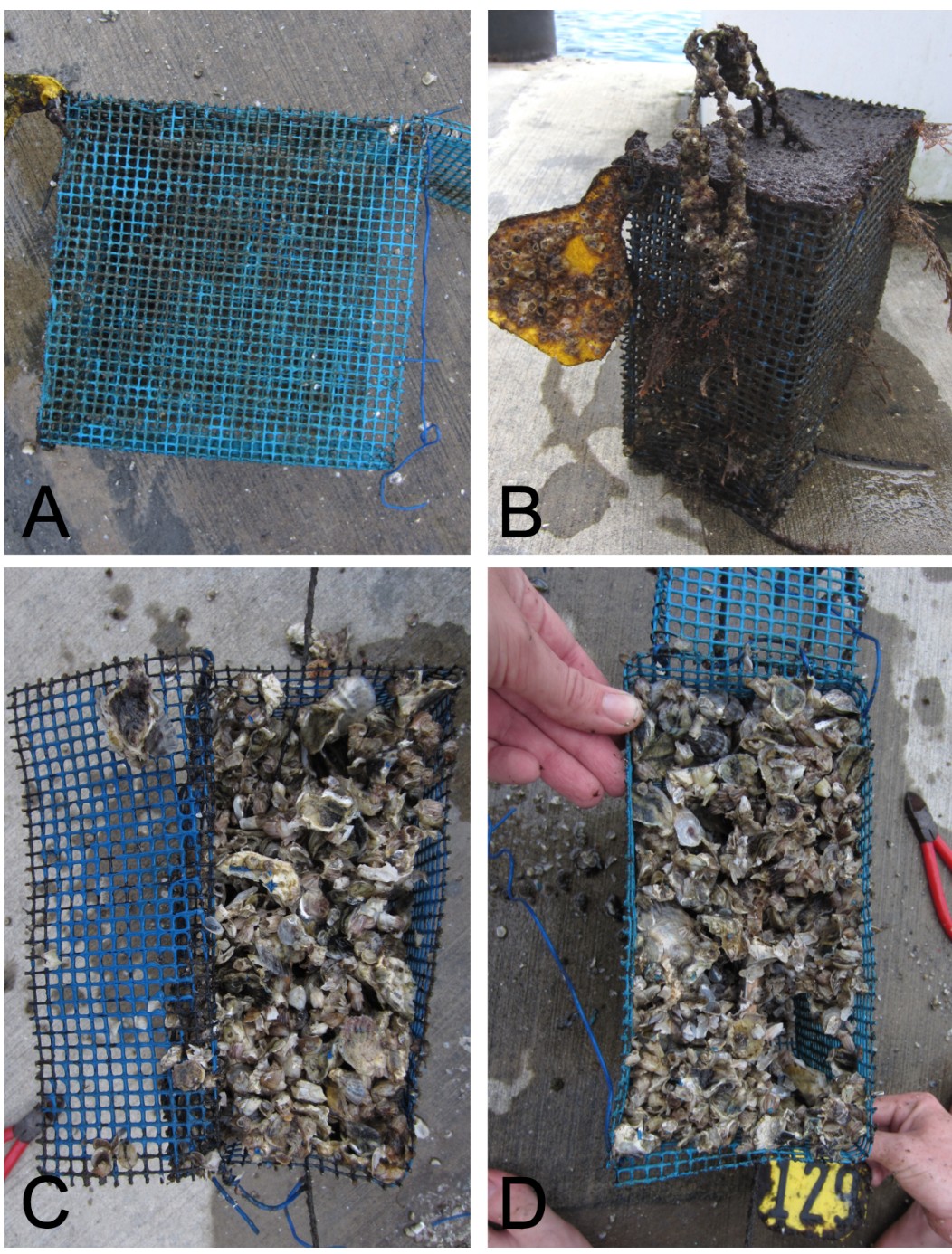

**Figure 5 Photographs of post experiment exclusion cages.** Experimental cages after the additional two-month deployment following the primary six-week experiment. Biofouling on cages with copper-based paints remained low (A) while turf algae tended to cover those treated with ecofriendly paints (B). Left unattended, continued settlement to and growth on tiles filled the exclusion cages treated with 21% $Cu_2O$ (C) and 40% $Cu_2O$ paints (D).

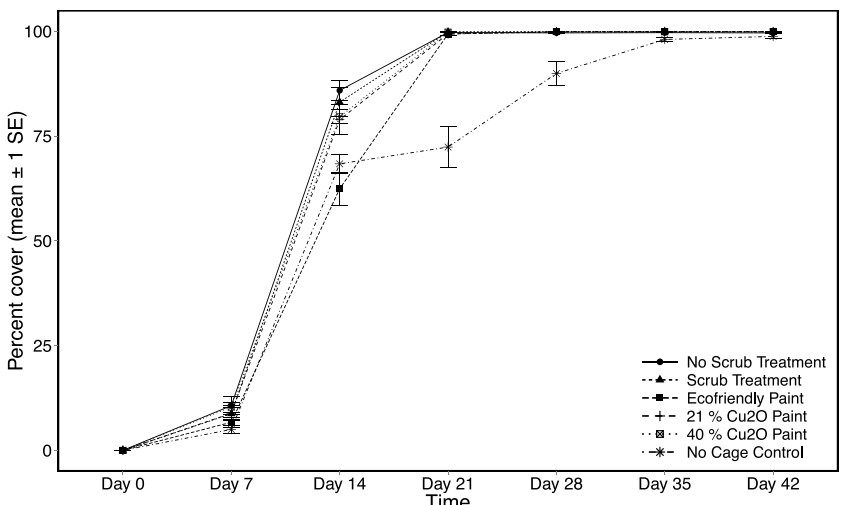

**Figure 6** **Settlement tile percent cover as a function of treatment and day.** Total percent cover (mean ± 1 SE) on settlement tiles measured weekly for six weeks across the six experimental treatments.

unsuitable for short-term settlement experiments. The ecofriendly paint composition includes chemicals such as titanium dioxide, zinc oxide, and zinc pyrithione.

Titanium dioxide (*Sharma*, *2009*) and zinc pyrithione (*Kobayashi & Okamura*, *2002*; *Karlsson & Eklund*, *2004*) have been shown to be toxic to a variety of marine organisms. Zinc oxide is typically used in conjunction with copper because it increases the toxicity of copper by 200-fold (*Watermann et al.*, *2005*). By itself, zinc oxide has a weak antifouling performance, but there is concern that its presence in paints may cause future water quality issues (*Watermann et al.*, *2005*; *Holman et al.*, *2011*). Although these chemicals are present in the ecofriendly paint we used, the current study cannot determine whether any of them elicited the observed effects on biomass and macro-community composition.

In addition to the differences in biomass and macro-community composition of settlement to the tiles, the paints tested had varying success in product performance on the cages. The copper-based painted cages performed well and retained negligible levels of biofouling throughout the experiment and additional deployment times. Conversely, the performance of the ecofriendly paint was relatively poor overall during the experiment. Specifically, during the additional two-month deployment, the ecofriendly painted cages became partially covered with turf algae dense enough on some parts of the cage to mask the blue paint color.

Although there is continued support and funding going into the development of ecofriendly and alternative paints, the compounds Econea[TM], cappasicin, and medetomidine, which are collectively referred to as "emerging" biocides, remain relatively unstudied (*Thomas & Brooks*, *2010*; *Guardiola et al.*, *2012*; *Gee et al.*, *2013*). Assuming that a reduction in biomass and a shift in macro-community composition is indicative of detrimental effects to the local environment, the results of this study justify further investigation into the ecological ramifications of Econea[TM] and other copper-free biocides.

The lack of local effects from the copper paints presented in this study contradicts previous research. For example, increased exposure to heavy metal pollution (i.e., International® Micron Extra 25–50% cuprous oxide) has been shown to alter macro-community composition and decrease native species densities, permitting the dominance of more tolerant, non-indigenous ones (*Piola & Johnston 2008*). *Canning-Clode et al.* (*2011*) found that as exposure to antifouling paint increased, community composition changed and the densities decreased for both native and non-indigenous species. However, it should be noted that *Canning-Clode et al.* (*2011*) used paint with much higher copper concentrations (i.e., Interlux® Ultra-Kote 76% cuprous oxide) than what was used in the present study. It is possible that the decreased levels of copper in the present study did not surpass organismal tolerances, thus the observed lack of local negative effects. Furthermore, *Canning-Clode et al.* (*2011*) exposed the marine communities to various copper concentrations by painting borders with different loads of antifouling paint around the biofouling communities. This direct contact, versus the 5 cm buffer used here, may also explain why macro-community compositions of the copper paint treatments did not differ from the scrubbed-cage treatment. It is also possible that disparity in deployment time may account for the differences between previous studies and the present study, since that by *Piola & Johnston* (*2008*) lasted seven months and *Canning-Clode et al.* (*2011*) lasted nine weeks.

The copper-based ablative paints were manufactured to last for approximately 12 months and require physical abrasion by water in high vessel velocities or manual removal by methods such as power washing for paint layers to ablate. It is likely that due to the short duration of the study and the location of the experiment (i.e., protected waters in a harbor), the paints did not experience forces strong enough to consistently ablate, thus the observed lack of an effect on the tile within the cage. However, small specs of the copper-based paints were occasionally observed on the tiles, yet the direct contact with the paint particulates did not affect the biomass and composition of the communities on tiles in the reduced-copper painted cages. Further studies can expand on the present study's focus (i.e., short-term settlement patterns and trends) to test the effects of reduced-copper paints on biological and/or ecological process beyond settlement. Although it was not tested in the present study, the potential of bioaccumulation of heavy metals may pose serious problems to studies that measure growth, survival, or other biological and ecological processes over longer periods of time that those measured here (*Bao et al.*, *2010*; *Qi et al.*, *2015*).

The observed effects on tiles in the no-cage and no-scrub treatments were expected. The distinct macro-community, biomass, and percent cover observed in the no-cage treatment can be attributed to the signs of grazing from local predators such as the common Sheepshead (*Archosargus probatocephalus*; JJ Walbaum). Additionally, the distinct macro-community and decreasing rate of biomass accumulation in the no-scrub treatment was likely due to biofouling buildup on the cage that subsequently obstructed water flow and settlement on the tile within it. While the no-cage treatment was intended to reflect a common experimental design used to separate pre- and post-settlement processes, the no-scrub treatment illustrated the importance of accounting for biofouling of cage materials. Addressing biofouling using the traditional approach of frequent manual removal

will likely continue to be the primary method employed by field scientists, but doing so is not always logistically feasible. The newly developed reduced-copper antifouling paint compositions, such as those used here, have the potential to act as a viable alternative for manual maintenance of infrastructures, such as cages.

## ACKNOWLEDGEMENTS

We thank Bill Wolf from Pettit Paint, who graciously donated the three antifouling paints to our research and provided feedback on product performance. Owen Stokes-Cawley and Matthew Farnum helped with fieldwork and data collection and Jonathan Grabowski provided helpful feedback and editorial assistance. This research was in partial fulfillment of a Master's degree conferred to ASJ from the Three Seas Program at Northeastern University.

### Funding

Supplies were purchased with Education and General (E&G) funds to CD Stallings from the University of South Florida. The funders had no role in study design, data collection and analysis, decision to publish, or preparation of the manuscript.

### Grant Disclosures

The following grant information was disclosed by the authors:
University of South Florida.

### Competing Interests

The authors declare there are no competing interests.

### Author Contributions

- Andrea S. Jerabek conceived and designed the experiments, performed the experiments, analyzed the data, wrote the paper, prepared figures and/or tables, reviewed drafts of the paper.
- Kara R. Wall conceived and designed the experiments, performed the experiments, reviewed drafts of the paper.
- Christopher D. Stallings conceived and designed the experiments, contributed reagents/materials/analysis tools, wrote the paper, reviewed drafts of the paper.

### Patent Disclosures

The following patent dependencies were disclosed by the authors:
Hydrocoat Eco Ablative Antifouling Paint®
CPP Ablative Antifouling Paint®
Horizons Ablative Antifouling Bottom Paint®
Econea®.

## Data Availability

The raw data has been supplied as a Supplemental Dataset.

## Supplemental Information

Supplemental information for this article can be found online at http://dx.doi.org/10.7717/peerj.2213#supplemental-information.

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
