# Peer review of "­A practical application of reduced-copper antifouling paint in marine biological research"

_PeerJ, doi:10.7717/peerj.2213_

## Round 0.1 · original submission · Major Revisions

Dear Authors,

The reviewers found that your ms did not provide sufficient information in terms of how your work being done and showed substantial concerns about your ms. After going through your ms and their comments, I concur their comments and concerns. Overall, the ms can not be accepted at this present form and required substantial revision. Although I rated this ms as a major revision but if you can't make substantial improvement, this will lead to decline in the end.

Reviewer 1 ·

Basic reporting

No Comments

Experimental design

In M&M, the “percent cover” were measured based on the images. The calculation for the respective “percent cover” for each type of fouling organisms was not clearly described. Although author explained in M&M, “After the adjustment, the fouling organisms were visually distinct from the tile”, the distinct method for organisms from each other in images was absent.

In the results of “Community composition on tiles”, the appearing time of each type of organisms in a table (Like a schedule) should be better than only mentioning in the Text.

The statistical analysis results are very important for data of community composition and others, but missed in figures. It is suggested to add markers in the figures to notice the statistical differences.

Validity of the findings

No Comments

Additional comments

The applications of toxic antifouling paints on the surfaces of marine scientific instruments and facilities should be cautious, because they might bring unexpected side effects. For marine biologists, such applications usually will be extra cautious. If they are used for biofouling and antifouling field tests, the possible combined effects of testing components with Copper should be considered. In addition, the commercial antifouling paints contain many other ingredients. For example, the Econea-based paint normally contains an arylpyrrole compound, which was not fully studied yet, in addition to Zinc and so on. For other purposes, the possible biologic accumulations of Copper might be a problem. Therefore, the author should clearly explain such painted predator exclusion cages could be used for what purpose and circumstance in marine biological research.

Reviewer 2 ·

Basic reporting

No Comments

Experimental design

The measurement of community composition and biomass of macrofouling should be provided in detail.

Validity of the findings

No Comments

Additional comments

The authors described their research on antifouling performance of three paints to find a substitute for manual experimental cage maintenance. The manuscript is generally of interest to readers in the field of Biofouling or Aquaculture. In general, this paper was not well written, and I think re-submission or rejection may be appropriate before publication in PeerJ.

Below, general and specific comments were provided.
General comments:
It is difficult to follow the methods, and the authors need clarify the measurement of community composition and biomass of macrofouling.

The discussion section needs to be reinforced and reorganized.

Specific comments:
Abstract
The abstract needs to be rewritten to make it clear. Less description of background and more detail of results may be appropriate.

Materials and methods:
More information on the measurement of community composition and biomass of macrofouling organisms should be provided.

Results
Line 145-158: The data analysis needs to be provided to compare the antifouling performance of these treatments because this is a key result.

Figure 2: SEM should be replaced with “SE”.
Good quality of Figure 4 and 6 should be provided due to difficulty in understanding for the readers.

Line 172-176: rewrite this sentence to make clear.

Discussion:
The authors should compare their work with previous research, not only provide some speculation in Line 216-227.
Generally, this section needs to be rewritten. The authors need focus on the difference between paints and control, between reduced-copper paints and copper-free paint.

---

## Round 0.2 · Minor Revisions

Please note that additional minor comments have been made by Reviewer 2. They should not be difficult to address, and we loo forward to your revision.

Reviewer 1 ·

Basic reporting

No Comments

Experimental design

The M&M is clearly now showing experimental design.

Validity of the findings

No comments

Additional comments

In general, the revised MS is much improved in writing as well as in organizing. I suggest acceptance for its publication.

Reviewer 2 ·

Basic reporting

No comment

Experimental design

No comment

Validity of the findings

No comment

Additional comments

The research topic is of great interest in the new coating applications. The paper has been much improved, but still has weaknesses specifically with regards to scientific writing. After minor revision, this manuscript can be accepted in PeerJ.

Specific comments:
Abstract
Lines 7-8: Rewrite this sentence “Six treatment levels were tested, three with and three without antifouling paints”.
Line 12: Replace “restrict” with “reduce”.
Lines 17-19: “composition” revised to “composition of marcofouling community”.

M&M
Replace “community composition” with “macrofouling community composition” in the whole manuscript. Ex: Line 47, 122, 154 and etc.

Results
Line 169-170: Delete “By the end of the experiment”, and can use “by day” in the manuscript.
Lines177-178: Rewrite this sentence. Delete “By the completion of the experiment”. What is the meaning of “clear patterns”?
Line 180: replace “greatest” with “highest”; Line 182, 183, 184: replace “greater” with “higher”.
Lines 204: Rewrite “At that time”. Rewrite this sentence “percent cover on tiles inside cages painted with the ecofriendly paint was not different from the no-cage treatment”. e.g. “No significant difference of percent cover on tiles was observed between ecofriendly paint and the no-cage treatment”.
Line 207: delete “had”; replace “rendering” with an appropriate word.
Line 209: replace “persisted” with an appropriate word.

Discussion
Lines 273: replace “ours” with “the present study”.

---

## Round 0.3 · accepted · Accept

The revised has addressed the reviewer's concern adequately and is accepted for publication.